

# Effects of social isolation and re-socialization on cognition and ADAR1 (p110) expression in mice

Wei Chen[1,*], Dong An[1,*], Hong Xu[2,*], Xiaoxin Cheng[1], Shiwei Wang[3], Weizhi Yu[1], Deqin Yu[1], Dan Zhao[1], Yiping Sun[1], Wuguo Deng[4], Yiyuan Tang[5] and Shengming Yin[1]

[1] College of Basic Medical Sciences, Dalian Medical University, Dalian, China
[2] Department of Physiology Laboratory, Dalian Medical University, Dalian, China
[3] Menzies Research Institute, University of Tasmania, Tasmania, Australia
[4] Institute of Cancer Stem Cell, Dalian Medical University, Dalian, China
[5] Department of Psychological Sciences, Texas Tech University, Lubbock, United States
[*] These authors contributed equally to this work.

Corresponding author
Shengming Yin,
dlshengming@163.com

## ABSTRACT

It has been reported that social isolation stress could be a key factor that leads to cognitive deficit for both humans and rodent models. However, detailed mechanisms are not yet clear. ADAR1 (Adenosine deaminase acting on RNA) is an enzyme involved in RNA editing that has a close relation to cognitive function. We have hypothesized that social isolation stress may impact the expression of ADAR1 in the brain of mice with cognitive deficit. To test our hypothesis, we evaluated the cognition ability of mice isolated for different durations (2, 4, and 8 weeks) using object recognition and object location tests; we also measured ADAR1 expression in hippocampus and cortex using immunohistochemistry and western blot. Our study showed that social isolation stress induced spatial and non-spatial cognition deficits of the tested mice. In addition, social isolation significantly increased both the immunoreactivity and protein expression of ADAR1 (p110) in the hippocampus and frontal cortex. Furthermore, re-socialization could not only recover the cognition deficits, but also bring ADAR1 (p110) immunoreactivity of hippocampus and frontal cortex, as well as ADAR1 (p110) protein expression of hippocampus back to the normal level for the isolated mice in adolescence. In conclusion, social isolation stress significantly increases ADAR1 (p110) expression in the hippocampus and frontal cortex of the mice with cognitive deficit. This finding may open a window to better understand the reasons (e.g., epigenetic change) that are responsible for social isolation-induced cognitive deficit and help the development of novel therapies for the resulted diseases.

## INTRODUCTION

Social isolation, a kind of psychosocial stressor (*O'Keefe et al., 2014*; *Barratt, Shaban & Moyle, 2011*), is defined as an objective reduced social contact (*Khodaie et al., 2015*). To understand the negative impact of social isolation, researchers have investigated how social isolation affects the behavioral, psychological, and physiological mechanisms of

both human beings and animals (*Valtorta et al., 2016*). It has been reported that early adverse social events can significantly damage the development and maturation of brains, leading to morphological and functional abnormalities of the central nervous system (*Murai et al., 2007*). The syndrome resulted from social isolation has been found in rats (*Hatch et al., 1965*), mice (*Valzelli, 1973*), as well as monkeys (*Bowden & McKinney, 1972*). In addition, social isolation leads to behavior changes of adult rodents, with similar symptoms for patients experiencing from neuropsychiatric diseases, such as attention deficit hyperactivity disorder, obsessive-compulsive disorder, autism, schizophrenia, and depression (*Koike et al., 2009*). Investigations on both humans (*Grant, Hamer & Steptoe, 2009*) and rodent models (*Fone & Porkess, 2008*) indicate that social isolation can lead to cognitive dysfunction (*Yusufishaq & Rosenkranz, 2013*). Many studies have explored the possible reasons that are responsible for social isolation stress-induced cognitive deficit, including the alterations of glutamate receptors (*Araki et al., 2014*; *Meffre et al., 2012*), neurotransmitters (*Baarendse et al., 2013*; *Hellemans, Nobrega & Olmstead, 2005*), ion channels (*Quan et al., 2010*), Neural cell adhesion molecule (*Pereda-Pérez et al., 2013*; *Djordjevic et al., 2012*), and the hypothalamo-pituitary-adrenal (HPA) axis (*Sandstrom & Hart, 2005*; *Pisu et al., 2016*; *Haj-Mirzaian et al., 2016*). However, detailed mechanism is still not clear. So far, little is known whether there is any direct link between social isolation and ADAR1 expression.

ADAR1 (Gene ID: ADAR) is widely distributed in the central nervous system (*Yang et al., 2003*) and belongs to ADAR family. ADAR1 proteins possess two types of nucleic acid binding domains. Three copies of a double-stranded RNA binding motif are present in the central region of the p110 and p150 proteins (*Patterson & Samuel, 1995*; *Liu & Samuel, 1996*; *Fierro-Monti & Mathews, 2000*). The dsRNA binding motifs found in ADAR1 proteins are similar to the prototypical dsRNA binding motif discovered initially in protein kinase R (*Toth et al., 2006*). Both p150 and p110 are active enzymes that catalyze the C6 deamination of adenosine in duplex RNA structures. Genetic disruption of the mouse ADAR1 gene by knocking out the expression of both p150 and p110 ADAR1 proteins results in embryonic lethality (*Hartner et al., 2004*; *Wang et al., 2004*; *George, John & Samuel, 2014*).

ADAR1-mediated RNA editing has been shown to affect a number of biological responses including virus growth and persistence, cell proliferation, neurotransmitters, and innate immune responses (*Bombail et al., 2014*; *Cattenoz et al., 2013*; *Mattick & Mehler, 2008*). Moreover, pre-mRNA of 5-HT2C receptor (*Schirle et al., 2010*; *Schmauss et al., 2010*), AMPA receptor, GABA receptor, and KV1.1 potassium channel catalyzed by ADAR family have been reported to have closed relation to cognition (*Bombail et al., 2014*; *Cattenoz et al., 2013*; *Mattick & Mehler, 2008*). In addition, some study indicates that social isolation can disturb the normal neurotransmitter and innate immune responses (*Okada et al., 2015*) and induce cognitive deficit (*Yusufishaq & Rosenkranz, 2013*). So we have hypothesized that there may be a direct link between social isolation and ADAR1 expression.

In order to test our hypothesis, post-weaning Kunming (KM) mice were isolated for 2, 4, and 8 weeks, respectively. Object recognition and object location tests were used to evaluate the cognition of the isolated mice. Moreover, ADAR1 (p110) expression in frontal

cortex and hippocampus was measured using immunohistochemistry and western blot. Furthermore, the changes of ADAR1 (p110) expression in hippocampus and frontal cortex for behavioral deficit recovery were evaluated for the mice with re-socialization. Our results have demonstrated that cognitive deficit resulted from social isolation is related to ADAR1 (p110) over expression in the brain. This finding may open a window to better understand the reasons that are responsible for social isolation-induced cognitive deficit and help the development of novel therapies (e.g., epigenetics) for the resulted disorders from social isolation.

## MATERIALS AND METHODS

### Animals

Male KM mice (15 ± 5 g) at the age of 21 days old were purchased from Laboratory Center of Dalian Medical University (ID: 0003746). They were housed in the plastic cage (Beijing Heli Technology Development Co. Ltd. China, 290 × 178 × 160 mm) with 5 mice for each cage reared in conditions of 21 ± 1 °C, 55 ± 5% humidity, and 12-h rhythm of day/night cycle. The mice were fed with food and water ad libitum. They were divided into seven groups randomly with 10 mice for each group, as shown in Fig. 1. The mice were housed individually for 2, 4, and 8 weeks respectively. They were labeled as SI 2W group (isolation for 2 weeks), SI 4W group (isolation for 4 weeks), and SI 8W group (isolation for 8 weeks). In addition, recovery of behavioral deficit by re-socialization (rearing with their littermates at adolescence period) was also examined. The re-socialization mice were labeled as SI 2WR group (re-socialization for 2 weeks after isolation for 2 weeks). Control groups were age matched group-housed mice (C2W, C4W, and C8W). All experimental procedures were approved by the Tab of Animal Experimental Ethical Inspection, Number: L20140021.

### Methods

#### Object Recognition Test (ORT)

The minor modified protocol (Fig. 2) as published in the literature (*Võikar et al., 2005*; *Van Hagen et al., 2015*; *Cechella et al., 2014*) was used in this study to measure the non-spatial cognitive ability. The test was performed in the behavior procedure room using the behavior observation apparatus (XR-XX117, Shanghai Xinruan Technology Co. Ltd. China). The observed box was 40 × 40 × 35 cm. Woody block A and B (Black, 5 × 5 × 5 cm) and block C (Black and white pattern, 5 × 5 × 5 cm) were used as recognition objects, with each object heavy enough to be stable. The process for ORT included both sample trial and test trial. Firstly, the mouse was trained to be in the empty box for 5 min to acclimate to the new environment. Then, during the sample trial, object A and object B were placed oppositely with a distance of 14 cm between them. The exploring time for each mouse was 5 min. The mouse was placed in the middle of two objects in the test. After the sample trial, the mouse returned back to its home cage for 4 h. During the test trial, object B was replaced with object C that was a novel one for the mouse. After each trial, the objects and the box were cleaned in order to avoid olfactory cues using 75% ethanol. The behavior of the mouse was recorded by videotapes.

The discrimination index (DI) was calculated as follows: $DI = (Tn - Tf)/(Tf + Tn)$: $Tn$ and $Tf$ were the time taken for the mouse to recognize a new or familiar object, respectively.

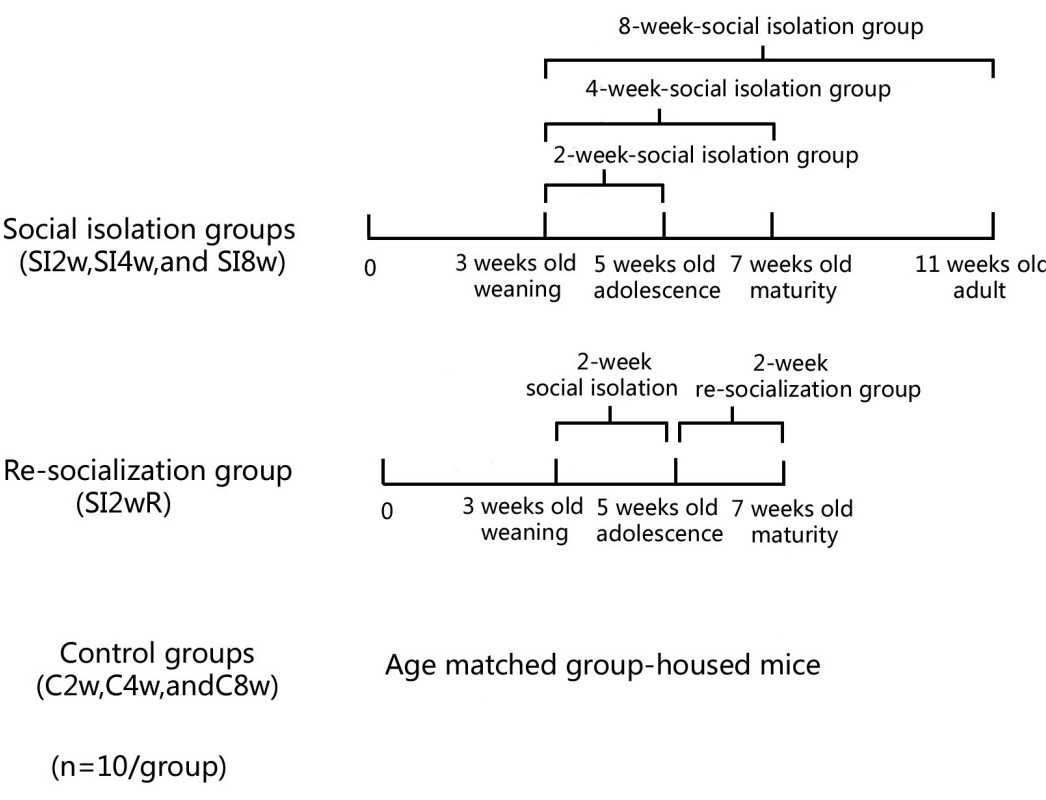

**Figure 1** **Treatment of divided mice groups with isolation and re-socialization.** The mice were housed individually for 2, 4, and 8 weeks respectively. They were labeled as SI2W group (isolation for 2 weeks), SI 4W group (isolation for 4 weeks), and SI 8W group (isolation for 8 weeks). In addition, recovery of behavioral deficit by re- socialization (rearing with their littermates at adolescence period) was also examined. The re-socialization mice were labeled as SI 2WR group (re-socialization for 2 weeks after isolation for 2 weeks). Control groups were the age matched group-housed mice (C 2W, C 4W, and C 8W).

### Object Location Test (OLT)

The performance of OLT was similar with that of ORT (*Võikar et al., 2005*; *Van Hagen et al., 2015*; *Cechella et al., 2014*). This test is used to measure the spatial cognitive ability. Object A and B used in this study were the same as those used in the ORT. The acclimation was carried out in the same way as that in the ORT. In the sample trial, object A and B were put in the same location as that in the ORT. The mouse explored each object for 5 min and then was returned to the home cage for 4 h. After that, in the test trial, object B was moved to the opposite direction toward the object A, then the mouse was left to explore object A and object B with a novel location for 5 min. The behavior of the mouse was recorded in the same way as that used in ORT.

### Immunohistochemistry staining

The mice were injected with 4% chloral hydrate for anesthesia (400 mg/kg, i.p.), followed by perfused transcardially with 1% and 4% paraformaldehyde (PFA) respectively. After that, the mice brains were fixed with PFA for 24 h at 4 °C and then cryoprotected in 20% sucrose at 4 °C overnight. Subsequently the brains were sliced with a cryostat with 16 μm thick. The slices were treated with phosphate-buffered saline (PBS) for three times with

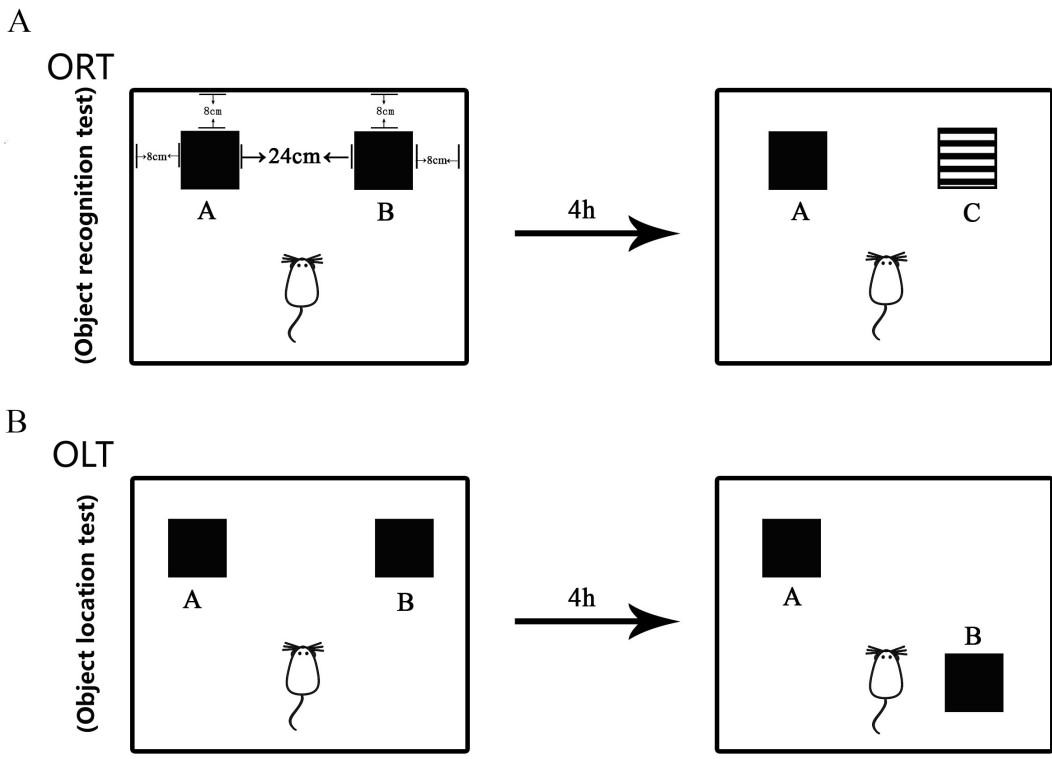

**Figure 2** **Diagram of Object Recognition Test and Object Location Test.** (A) Object Recognition Test. The diagram shows the apparatus and recognition objects used in this test. Woody block A and B (Black, $5 \times 5 \times 5$ cm) and block C (Black and white pattern, $5 \times 5 \times 5$ cm) were used as recognition objects. Objects A and B were identical. The process for ORT included both sample trial and test trial. During the sample trial, object A and object B were placed oppositely with a distance of 14 cm between them. In the test trial, object B was replaced by novel object C. (B) Object Location Test. The apparatus and objects A and B were the same as that used in the ORT. In the sample trial, objects A and B were put in the same location as that in the ORT. In the test trial, object B was moved to be located at the opposite direction toward the object A.

10 min for each time and then were incubated in 1% bovine serum albumin. After that, the slices were incubated with primary antibody of ADAR1-Ab (p110) (1:100, Proteintech, USA) overnight at 4 °C. Then the sections were washed with PBS and were cultivated with biotinylated second antibody (ZSJQ-BIO Company, China) at room temperature for 1.5 h. After washing with PBS, the slices were treated with avidin-biotin complex at room temperature for 2 h. The positive expression of ADAR1 (p110) was visualized with diaminobenzidine (DAB) for the detection. Negative control slices were incubated with PBS without antibody, followed by stained with 1% thionine (sigma), a specific staining of nissl body of the neuron. Image analysis for quantification of the results was performed.

### Western blot

Protein of frontal cortex and hippocampus was extracted by using the extraction kit (Keygen Biotech, China). The protein concentration was assessed using the BCA protein assay kit (Keygen Biotech, China). Proteins (30 µg per sample) were denatured and then loaded on 7.5% sodium dodecyl sulfate-polyacrylamide (SDS) gel. After that, proteins were

transferred to polyvinyl diflouride (PVDF) membranes blocked for 1 h with 5% bovine serum albumin and then immunoblotted with the primary antibody (ADAR1-Ab-p110, 1:1,000; Proteintech, Chicago, IL, USA). Subsequently, membranes were washed with tris-buffered saline with Tween 20 (TBST) and incubated with horseradish peroxidase-labeled secondary antibody (anti-goat 1:5,000; ZSJQ-BIO Company, Beijing, China) for 2 h at room temperature in the dark. The Infrared band signals were detected and quantified using BIO-RAD (Hercules, CA, USA) gel analysis software. Membranes were then stripped using stripping buffer, washed in TBST, and probed with GADPH-Ab (1:1,000, Beyotime Company, China). After washing with TBST, membranes were incubated with horseradish peroxidase-labeled secondary antibody (anti-mouse, 1:5,000; ZSJQ-BIO Company, Beijing, China) and then were detected. ADAR1 (p110) protein expression was normalized by internal control-GADPH.

### Statistical analysis

Graph-Pad Prism (GraphPad Software Inc.) and SPSS 21.0 were used to analyze statistically. All data were expressed as the mean $\pm$ SEM and were analyzed by two-way ANOVA followed by Tukey's post hoc testing. $T$ test was used to analyze the variance between matched social isolation group and control group; two-way ANOVA was used to understand whether there is an interaction between social isolation and age level (two independent variables) on cognitive function (dependent variable) among mice. Post-hoc tests (or post-hoc comparison tests) are used at the second stage of the analysis of variance (ANOVA) or multiple analyses of variance (MANOVA) if the null hypothesis is rejected. In our study, the values of isolation 2 weeks, 4 weeks, and 8 weeks were analyzed as multiple analyses of variance, as well as the values of control 2 weeks, 4 weeks, and 8 weeks. ANOVA was used to analyze the differences among groups. $P < 0.05$ was considered statistically significance.

## RESULTS

### Decreased DI of cognition by social isolation and its recovery by re-socialization

In the ORT and OLT, the DI of the mice isolated for 2, 4, and 8 weeks significantly decreased as compared to the age matched group-housed mice respectively. The data was shown as follows: (ORT: (2 weeks control group: $0.30 \pm 0.03$; 2 weeks isolation group: $-0.01 \pm 0.02$; $p = 0.005$); (4 weeks control group: $0.12 \pm 0.01$; 4 weeks isolation group: $-0.11 \pm 0.02$; $p = 0.005$); (8 weeks control group: $0.01 \pm 0.03$; 8 weeks isolation group: $-0.25 \pm 0.02$; $p = 0.020$); OLT: (2 weeks control group: $0.33 \pm 0.25$; 2 weeks isolation group: $-0.17 \pm 0.02$; $p = 0.003$); (4 weeks control group: $0.40 \pm 0.03$; 4 weeks isolation group: $-0.30 \pm 0.02$; $p = 0.000$); (8 weeks control group: $0.27 \pm 0.04$; 8 weeks isolation group: $-0.31 \pm 0.02$; $p = 0.002$)). The decreased DI showed the decreased spatial and non-spatial cognition ability for the mice isolated with 2, 4, and 8 weeks respectively. Moreover, no obvious difference was observed between the re-socialization group (SI2WR) and the control group (C4W). This result (Fig. 3) suggested that social isolation stress induced abnormal spatial and non-spatial cognition abilities, which however, could be recovered by re-socialization.

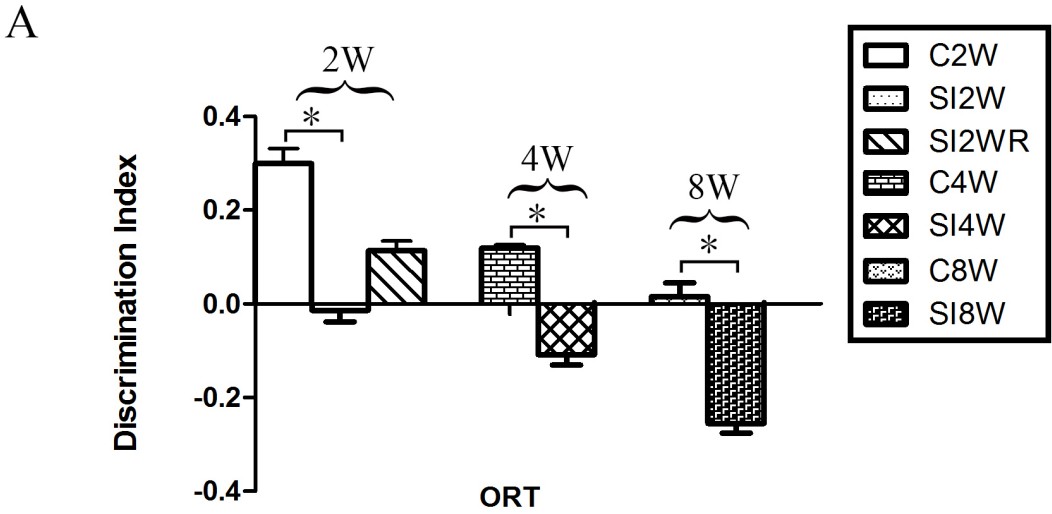

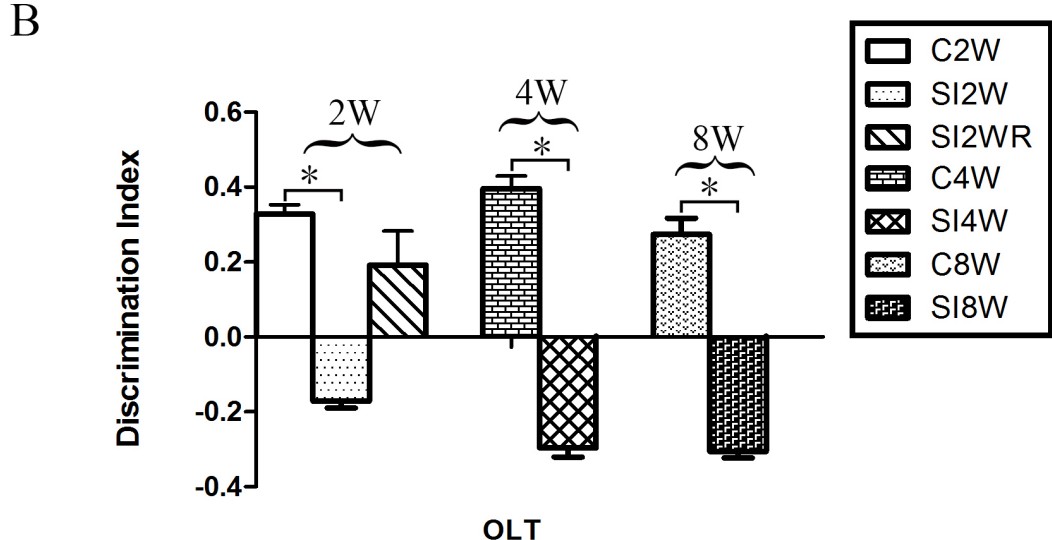

**Figure 3** **Decreased DI of spatial and non-spatial cognition in social isolation mice and its recovery by re-socialization.** Compared to the control group (C2W), 2 weeks social isolation (SI2W) resulted in decreased discrimination index (DI) of spatial and non-spatial cognition. Similarly, 4 and 8 weeks social isolation also decreased values of DI (SI4W vs. C4W and SI8W vs. C8W). Re-socialization (SI2WR) recovered the decreased DI of isolated mice (SI 4W). (A) results of ORT, (B) results of OLT. Data is presented as mean ± SEM ($n = 10$/group). *$P < 0.05$ (C2W vs. SI2W, C4W vs. SI4W, and C8W vs. SI 8W).

## Increased ADAR1 (p110) immunoreactivity by social isolation and its recovery by re-socialization

ADAR1 (p110) immunoreactivity was predominantly detected in frontal cortex and hippocampus in both control and social isolation mice (Fig. 4). In frontal cortex, ADAR1 (p110) immunoreactivity-positive signals were detected in only a part of nissl staining cells,

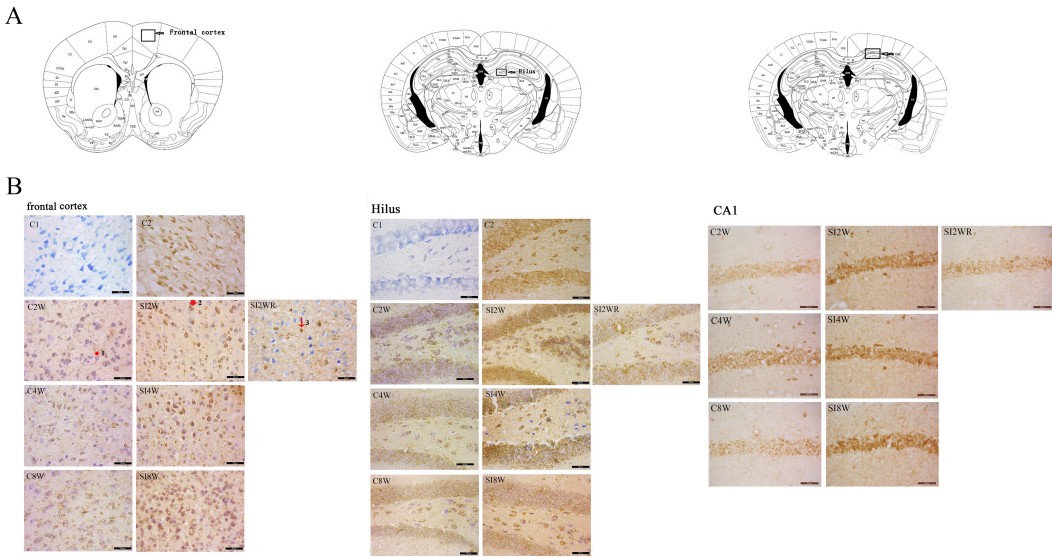

**Figure 4** **Increased ADAR1 (p110) immunoreactivity in frontal cortex and hippocampus of social isolated mice and its recovery by re-socialization.** ADAR1 (p110) immunoreactivity was predominantly detected in frontal cortex and hippocampus of control and social isolation mice, in the meanwhile, the number of detectable ADAR1 (p110) immunoreactivity positive signals significantly increased in the social isolation stress groups as shown in Fig. 4, compared to age matched group-housed control mice. In frontal cortex, the ADAR1 (p110) immunoreactivity positive signals displayed almost all layers from molecular layer to multiform layer. In hippocampus, ADAR1 (p110) positive signals widely distributed in CA1, dentate gyrus, and hilus. Re-socialization at adolescence recovered the increased ADAR1 (p110) immunoreactivity in frontal cortex and hippocampus of the isolated mice (Fig. 4). (A) The brain regions were analyzed on the basis of mice brain atlas of *Paxinos & Franklin (1997)*. Black boxes represented the brain regions magnified and presented as following Fig. 4B. Scale bar = 50 μm. (B) Double staining was performed using DAB staining and 1% thionine—a specific staining for nissl body of cell. ADAR1-Ab (p110) was used for marking ADAR1 (p110) immunoreactivity-positive signals. Arrow2 represented ADAR1 (p110) immunoreactivity positive signals in nissl staining cell, arrow 1 represented nissl staining cell without ADAR1 (p110) immunoreactivity positive signals, and arrow 3 represented ADAR1 (p110) immunoreactivity positive signals expressed in negative nissl staining region.

as seen in arrow 2 in Fig. 4, in the meanwhile, a partial of nissl staining cells did not show ADAR1 (p110) immunoreactivity-positive signals, as shown in arrow 1. Interestingly, a partial of ADAR1 (p110) immunoreactivity-positive signals were not detected in the nissl staining cells, as shown in arrow 3. The above distribution character of ADAR1 (p110) suggest that ADAR1 (p110) expressed in not only some specific neurons but also glia cells may be involved in the mechanisms of isolation-induced cognitive deficit, however, the more evidence need to be provided to test that, which will be studied further in our future work. The ADAR1 (p110) immunoreactivity-positive signals displayed almost all layers from molecular layer to multiform layer of frontal cortex. We counted and analyzed ADAR1 (p110) immunoreactivity-positive signals in layer 5 with closed relation to cognition. In hippocampus, ADAR1 (p110) immunoreactivity-positive signals distributed widely in CA1, dentate gyrus, and hilus (Fig. 4). Compared to the relative control groups, the immunoreactivity of ADAR1 (p110) significantly increased in hippocampus and frontal cortex of the mice isolated for 2, 4, and 8 weeks, respectively. As shown in Fig. 5, the

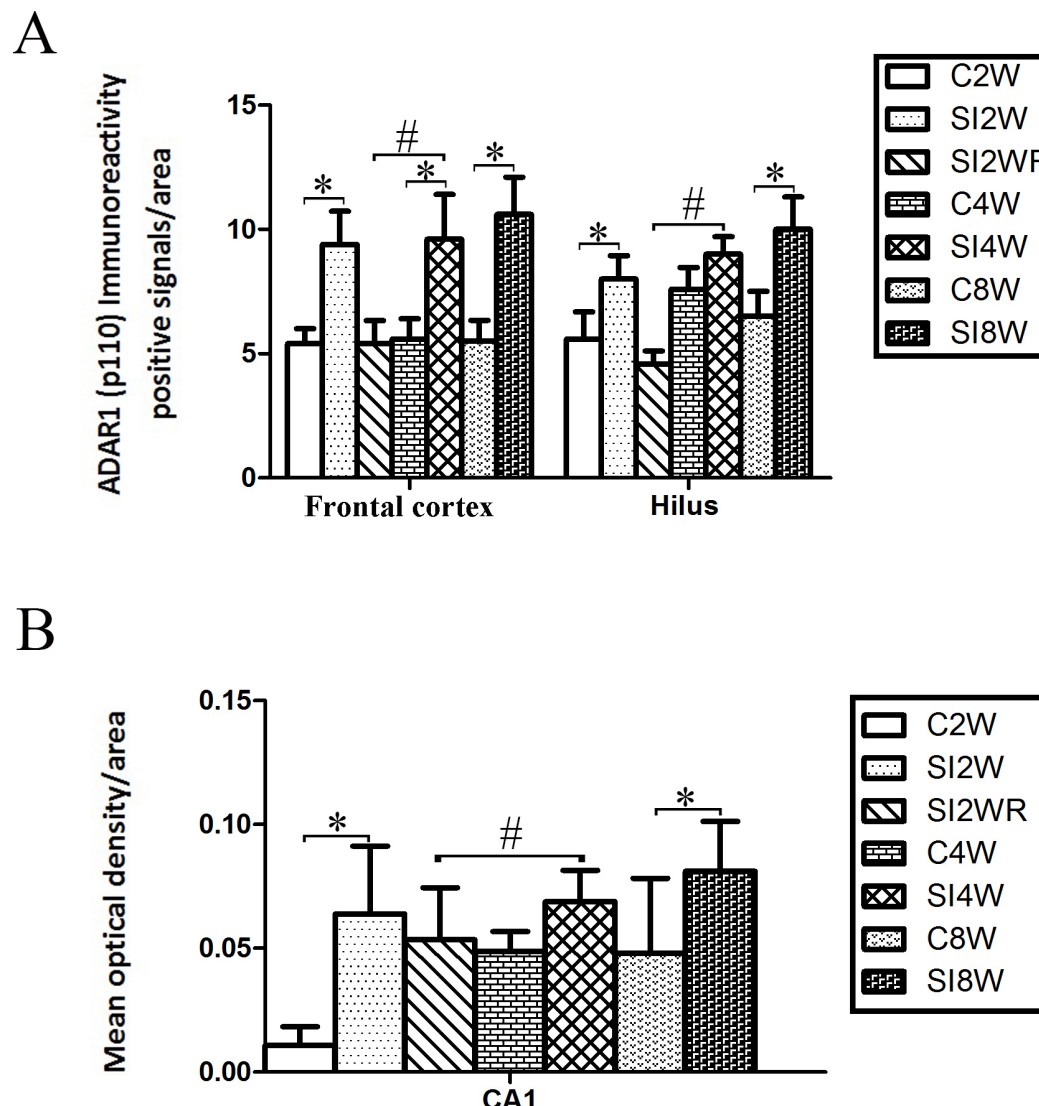

**Figure 5** **Increased ADAR1 (p110) immunoreactivity-positive signals in frontal cortex and hippocampus of the isolated mice by statistical analysis.** (A) The number of ADAR1 (p110) immunoreactivity positive signals/area in frontal cortex significantly increased in the mice isolated for 2, 4, and 8 weeks, compared to age matched group-housed control mice (C2W vs. SI2W, C4W vs. SI4W, and C8W vs. SI8W). In the meanwhile, the number of ADAR1 (p110) immunoreactivity positive signals/area in hilus significantly increased in the mice isolated for 2 and 8 weeks, compared to age matched group-housed control mice (C2W vs. SI2W and C8W vs. SI8W). (B) The mean optical density/area of ADAR1 (p110) immunoreactivity positive signals in CA1 significantly increased in the mice isolated for 2 and 8 weeks, compared to age matched group-housed control mice (C2W vs. SI2W, C8W vs. SI8W). Re-socialization recovered the increased ADAR1 (p110) immunoreactivity in frontal cortex and hippocampus of the isolated mice (SI4W vs. SI2WR). The number of ADAR1 (p110) immunoreactivity positive signals was counted in the sequential cutting sections. The sections analyzed in frontal cortex were from Bregma 1.18 mm for 4 sections, and the sections analyzed in hippocampus were from Bregma −2.18 mm for 4 sections (16 μm per section). The brain regions analyzed were focused on the internal pyramidal cell layer 5 of frontal cortex, as well as hilus and CA1 of hippocampus. The square analyzed was 10,000 μm$^2$. Data were expressed as the mean ± SEM and were analyzed by two-way ANOVA followed by Tukey's post hoc testing ($n = 5$/group). *$P < 0.05$ (C2W vs. SI2W, C4W vs. SI4W, and C8W vs. SI8W). #$P < 0.05$ (SI4W vs. SI2WR).
number of ADAR1 (p110) immunoreactivity-positive signals in frontal cortex of the mice isolated for 2, 4, and 8 weeks were significantly increased, compared to age matched group-housed mice. The data was shown as follows: (Number: (2 weeks control group: $5.40 \pm 0.60$; 2 weeks isolation group: $9.40 \pm 1.33$; $p = 0.02$); (4 weeks control group: $5.60 \pm 0.81$; 4 weeks isolation group: $9.60 \pm 1.81$; $p = 0.04$); (8 weeks control group: $5.50 \pm 0.85$; 8 weeks isolation group: $10.6 \pm 1.50$; $p = 0.01$)). However, the optical density of ADAR1 (p110) immunoreactivity-positive signals of CA1 and the number of ADAR1 (p110) immunoreactivity-positive signals of hilus were only increased obviously in the mice isolated for 2 and 8 weeks, compared to age matched group-housed mice. The data was shown as follows: (Number of hilus : (2 weeks control group: $5.60 \pm 1.08$; 2 weeks isolation group: $8.00 \pm 0.94$; $p = 0.04$); (8 weeks control group: $7.60 \pm 0.88$; 8 weeks isolation group: $10.00 \pm 1.31$; $p = 0.04$)) and (Optical density of CA1: (2 weeks control group: $0.01 \pm 0.02$; 2 weeks isolation group: $0.06 \pm 0.03$; $p = 0.04$); (8 weeks control group: $0.05 \pm 0.03$; 8 weeks isolation group: $0.08 \pm 0.02$; $p = 0.03$)). Furthermore, no obvious difference of number and optical density of ADAR1 (p110) positive immunoreactivity signals was found between the re-socialization group (SI2WR) and the control group (C4W), which suggested that re-socialization mice recovered the increased ADAR1 immunoreactivity in both frontal cortex and hippocampus of the isolated mice.

## Increased ADAR1 (p110) protein expression by social isolation and its recovery by re-socialization

Western blot results of ADAR1 (p110) were consistent with those of immunohistochemistry staining results mostly. The protein expressions of ADAR1 (p110) significantly increased in not only the frontal cortex but also the hippocampus of the mice isolated for 2, 4, and 8 weeks, compared to age matched group-housed mice, as shown in Figs. 6 and 7. The data was shown as follows: (the optical density of the ratio between ADAR1 (p110) and GADPH: frontal cortex: (2 weeks control group: $0.73 \pm 0.04$; 2 weeks isolation group: $1.02 \pm 0.12$; $p = 0.03$); (4 weeks control group: $0.25 \pm 0.08$; 4 weeks isolation group: $0.84 \pm 0.14$; $p = 0.04$); (8 weeks control group: $0.25 \pm 0.09$; 8 weeks isolation group: $0.72 \pm 0.12$; $p = 0.04$); Hippocampus: (2 weeks control group: $0.91 \pm 0.09$; 2 weeks isolation group: $1.41 \pm 0.28$; $p = 0.04$); (4 weeks control group: $0.65 \pm 0.19$; 4 weeks isolation group: $1.46 \pm 0.40$; $p = 0.04$); (8 weeks control group: $0.41 \pm 0.14$; 8 weeks isolation group: $0.71 \pm 0.13$; $p = 0.03$)). The above results suggested that social isolation increased ADAR1 (p110) protein expression. In addition, the protein expression of ADAR1 (p110) of SI2WR in hippocampus was no difference with that of age matched group-housed mice. The results suggested that re-socialization bring ADAR1 (p110) protein expression of hippocampus back to the normal level for the isolated mice in adolescence.

## DISCUSSION

In this study, we found that social isolation stress clearly induced spatial and non-spatial cognition deficits. In addition, social isolation significantly increased both the immunoreactivity and protein expression of ADAR1 (p110) in the hippocampus and frontal cortex of the tested mice. Furthermore, for the isolated mice in adolescence, re-socialization could

## frontal cortex

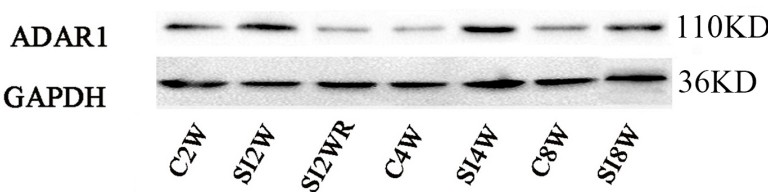

## hippocampus

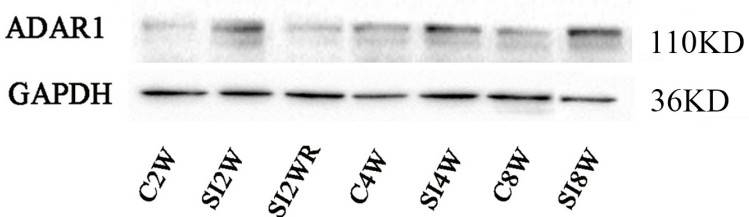

**Figure 6** **Increased ADAR1 (p110) protein expression in frontal cortex and hippocampus of isolated mice and its recovery by re-socialization.** The protein expression of ADAR1 (p110) significantly increased in the social isolation stress groups, as shown in Fig. 6, compared to age matched group-housed control mice. The detailed results were that ADAR1 (p110) protein expression increased in both frontal cortex and hippocampus of the mice isolated for 2, 4, and 8 weeks, in the meanwhile, re-socialization mice recovered the increased ADAR1 (p110) protein expression in the hippocampus of the isolated mice. The analyzation was shown in the following Fig. 7.

not only recover the cognition deficit, but also bring ADAR1 (p110) immunoreactivity of hippocampus and frontal cortex, as well as ADAR1 (p110) protein expression of hippocampus back to the normal level for the isolated mice in adolescence. Theses novel findings suggest that ADAR1 (p110) is related to isolation-induced cognitive deficit.

## Cognitive deficit induced by social isolation

Our findings demonstrated that social isolation resulted in spatial and non-spatial cognitive deficits in ORT and OLT, as illustrated in Fig. 3. These results agree with what has been reported in the literatures that social isolation can lead to cognitive dysfunction in rodent models (*Fone & Porkess, 2008*; *Dang et al., 2015*). However, mechanism is still not clear. The possible reasons responsible for the isolation-induced cognition deficit include the alterations of glutamate receptors (*Araki et al., 2014*; *Meffre et al., 2012*), neurotransmitters (*Baarendse et al., 2013*; *Hellemans, Nobrega & Olmstead, 2005*), ion channels (*Quan et al., 2010*), Neural cell adhesion molecules (*Pereda-Pérez et al., 2013*; *Watson et al., 2012*), and hypothalamo-pituitary-adrenal (HPA) axis (*Sandstrom & Hart, 2005*; *Gong et al., 2016*; *Watson et al., 2016*). The present study has demonstrated that ORT and OLT are the appropriate measures of cognitive deficit by social isolation.

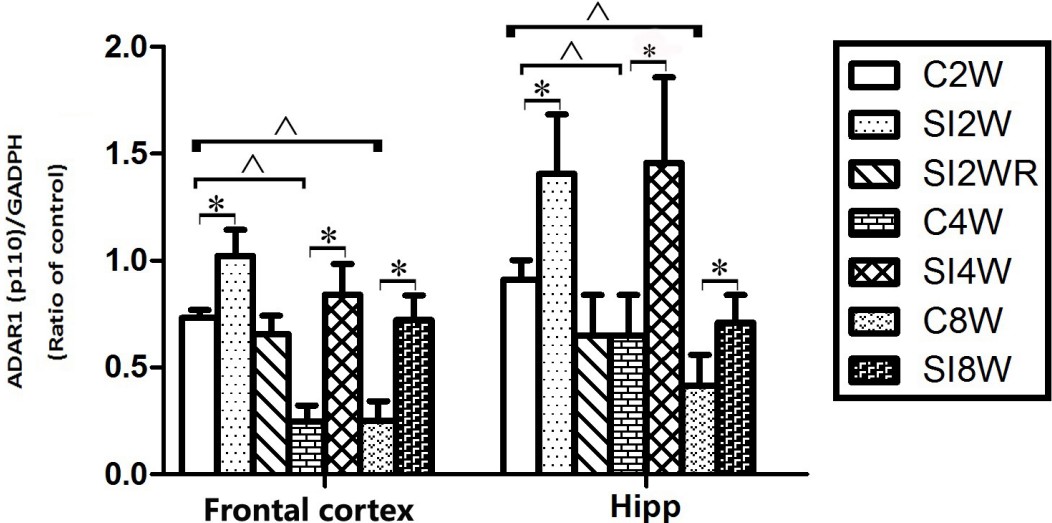

**Figure 7** **Increased ADAR1 (p110) protein expression in frontal cortex and hippocampus of isolated mice and its recovery by re-socialization in statistical analysis.** The ADAR1 (p110) protein expression increased in both frontal cortex and hippocampus of the mice isolated for 2, 4, and 8 weeks , compared to age matched group-housed control mice (SI2W vs. C2W, SI4W vs. C4W, and SI8W vs. C8W). In the meanwhile, re-socialization mice recovered the increased ADAR1 (p110) protein expression of the isolated mice in hippocampus to normal level (no different between C4W and SI2WR). Interestingly, the ADAR1 (p110) protein expression in both frontal cortex and hippocampus showed an age-dependent manner, which can be seen that ADAR1 (p110) protein expression of 11 weeks (C8W) and 7 weeks old mice (C4W) were less than that of 5 weeks old mice (C2W). ADAR1 (p110) protein expression was normalized by internal control—GADPH. The data was expressed as the mean ± SEM and analyzed by two-way ANOVA followed by Tukey's post hoc testing. *$P < 0.05$ (C2W vs. SI2W, C4W vs. SI4W, and C8W vs. SI8W), ^$P < 0.05$ (C2W vs. C4W and C2W vs. C8W).

## Increased ADAR1 (p110) by social isolation in the brain of mice with cognitive deficit

Although ADAR1 has been believed to have a close relation to cognitive function (*Bombail et al., 2014*; *Cattenoz et al., 2013*; *Mattick & Mehler, 2008*; *Schirle et al., 2010*; *Schmauss et al., 2010*), it is still unknown if there is any link between social isolation and ADAR1. ADAR1 is one of ADAR family and catalyzes the conversion process of adenosine to inosine (A-to-I) in post transcription level (*Buechel et al., 2014*). ADAR1 distributes widely in the central nervous system (*Liscovitch et al., 2014*; *Rybak-Wolf et al., 2015*). The expression level of ADAR1 mRNA is constant in the forebrain neocortex during postnatal development (*Schmauss et al., 2010*). However, little is known about the morphological distribution of ADAR1 in the brain of mice undergoing social isolation stress. We found that ADAR1 (p110) positive signals distributed widely on almost all layers (from molecular layer to multiform layer) of frontal cortex. Moreover, ADAR1 (p110) positive signals also expressed in CA1, dentate gyrus, and hilus of hippocampus (Fig. 4). For frontal cortex, we analyzed the change of ADAR1 (p110) immunoreactiviy-positive signals on pyramidal layer that plays an important role in cognitive function (*Elston, Benavides-Piccione & DeFelipe, 2001*). We found that the number of ADAR1 (p110) immunoreactivity-positive signals significantly increased in both frontal cortex and hippocampus for the isolated mice, compared with

that of age matched group-housed control mice, as shown in Fig. 4. It is important to note that both frontal cortex and hippocampus are vulnerable to social isolation stress (*Buechel et al., 2014*). According to the literature, ADAR-deficient mice exhibit defects in nervous system and decrease tolerance to stress (*Tseng et al., 2013*). It appears that ADAR1 (p110) expression is related to cognitive deficit induced by isolation stress.

## Recovered level of ADAR1 (p110) expression by re-socialization

In the past years, a number of methods have been reported to be able to alleviate the symptoms of cognitive deficit induced by social isolation, including re-socialization (*Maisonnette, Morato & Brandão, 1993*), electro-acupuncture (*Manni, Aloe & Fiore, 2009*), and treatment with drugs such as citalopram, cariprazine, aripiprazole, acetylcysteine (NAC), selective blockade of dopamine D3 receptors, and selective mGluR2/3 agonist (*Jones et al., 2011*). In this study, the effects of re-socialization on ADAR1 (p110) expression and the recovery of cognitive deficit were evaluated. It was interestingly found that re-socialization was effective to recover cognitive dysfunction and bring ADAR1 (p110) immunoreactivity of hippocampus & frontal cortex and ADAR1 (p110) protein expression of hippocampus back to the normal level for the isolated mice in adolescence. The reason for the recovery of cognitive deficit is possible that the mice in adolescence have strong neuro-plasticity and could be highly sensitive to re-socialization (*Forbes & Dahl, 2005*).

In conclusion, social isolation led to spatial and non-spatial cognition deficits in mice and impacted ADAR1 (p110) immunoreactivity and protein expression in hippocampus and frontal cortex. Moreover, re-socialization was effective to recover cognitive dysfunction and bring ADAR1 (p110) expression to the normal level for the isolated mice in adolescence. However, it is too early to make a conclusion that there is direct link between ADAR1 (p110) expression and cognitive deficit induced by social isolation, based on the evidence provided by this study so far. In the future, ADARs gene knock-out mice could be used to investigate how ADARs regulate RNA editing in social isolation-induced cognitive dysfunction. The related studies should be beneficial to the studies of social environment and body-mind healthy in human beings.

# ACKNOWLEDGEMENTS

We thank Dr Song Li for his technical support.

## Funding

This work was supported by grants from the National Natural Science Foundation of China (31201724). The funders had no role in study design, data collection and analysis, decision to publish, or preparation of the manuscript.

## Grant Disclosures

The following grant information was disclosed by the authors:
The National Natural Science Foundation of China: 31201724.

## Competing Interests

The authors declare there are no competing interests.

## Author Contributions

- Wei Chen conceived and designed the experiments, wrote the paper, prepared figures and/or tables.
- Dong An conceived and designed the experiments, prepared figures and/or tables.
- Hong Xu, Xiaoxin Cheng, Weizhi Yu and Deqin Yu performed the experiments.
- Shiwei Wang analyzed the data.
- Dan Zhao performed the experiments, prepared figures and/or tables.
- Yiping Sun and Wuguo Deng contributed reagents/materials/analysis tools.
- Yiyuan Tang contributed reagents/materials/analysis tools, advice the manuscript.
- Shengming Yin conceived and designed the experiments, wrote the paper, reviewed drafts of the paper.

## Animal Ethics

The following information was supplied relating to ethical approvals (i.e., approving body and any reference numbers):

The Tab of Animal Experimental Ethical Inspection—Number: L20140021.

## Data Availability

The raw data has been supplied as Data S1.

## Supplemental Information

Supplemental information for this article can be found online at http://dx.doi.org/10.7717/peerj.2306#supplemental-information.

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
