# Peer review of "Effects of social isolation and re-socialization on cognition and ADAR1 (p110) expression in mice"

_PeerJ, doi:10.7717/peerj.2306_

## Round 0.1 · original submission · Minor Revisions

· Academic Editor

Minor Revisions

This manuscript has been reviewed by three experts in the field. All of them found that this study is novel and the data are solid. However, there are a number of issues need to be addressed, especially the comments from the reviewer 3 about Figures and English editing. The authors are encouraged to make reversions and answer these comments one point by one point. The references citation and list should also follow PeerJ styles, especially for three authors.

·

Basic reporting

This manuscript is about the effects of social isolation stress on the cognitive function and the expression of ADAR1. They found that social isolation stress significantly increased ADAR1 expressions in the hippocampus and cortex, leading to cognitive deficits; further, they also evaluated recovery of the above effects by re-socialization in adolescence and found that re-socialization recovered not only the cognition deficits but also the increased ADAR1 expression in hippocampus and frontal cortex of the isolated mice in adolescence. Their interesting findings will be beneficial for clarifying the detailed mechanisms on social isolation stress leading to cognitive deficits.

I'd like to make the following suggestions:
1. The figure 1 includes figure A “the experiment design” and figure B and C “the introduction of OLT and ORT”. It will be better to make them to be separate parts. The figure should be re-designed.
2. The legend of Figure7 is unclearly, not mentioned “RG-enriched domain”, and the aim of showing the isoforms p110 and p150 of ADAR1 is also not clear, because there is no data of ADAR1 p150 in this paper.
3. Typo and grammatical errors need corrections.
4. The manuscript needs to be polished to make it more streamlined.

Experimental design

No Comments.

Validity of the findings

No Comments.

Reviewer 2 ·

Basic reporting

Summary:
The authors present interesting data about important roles of ADAR1 in cognitive deficits induced by social isolation. The authors report that the expression level of ADAR1 increased among mice suffered from social isolation and cognitive deficits which can be reversed by re-socialization. This research also helps build the possible case for the development of new drug curing cognitive deficits induced by social isolation. The manuscript is well written, has important messages, and should be of great interest to the readers. However, there are still some issues need to be revised. In general, I have some comments and suggestions to the authors listed below.

Experimental design

Abstract:
1. Need to add the impact of current research in this specific field.

Introduction:
1. The introduction provides a good, generalized background of the topic that quickly gives the reader a clear purpose of this study. However, to make the introduction
more substantial, the author may wish to provide several references to substantiate the possible underlying mechanisms beneath social isolation induced by cognitive deficits.
2. Please describe more about the influence of social isolation.
3. Add more description about properties of ADAR1 (eg. molecular structure).
4. Please cite references for “A-to-I RNA editing takes place in 5-hydroxytryptamine 2C receptor (5-HT2CR)…”.
5. “Center nervous system” should be “central nervous system”.
6. There is no link why you think ADAR1 connects to social isolation. Please specify.
7. It is necessary to add the summary, purpose and influence of your current research at the end of Introduction part.

Materials and methods:
Generally, the experimental apparatus is quite standard, and is appropriate for the study.
1. Please add more references for Object Recognition Test (ORT) and Object Location Test (OLT).
2. What is the post-hoc for one-way ANOVA?

Results:
Basically, the description of Results section is too simple, more detailed information should be added into Results section. It is not good that Results section looks like figure legends.

Discussion:
1. Please list current methods in order to alleviate the symptoms of cognitive deficits induced by social isolation if applicable.
2. There are no limitations of the study discussed. Think about what are the major limitations that should be discussed in the manuscript.
3. Need to write future directions of current research.
4. Check word spelling, word space and capital letters.

Figures
1. Please check carefully about spelling and format before submission. In Fig. 7, there should a space between “Diagram” and “for” in the figure title.

Validity of the findings

.

Comments for the author

My recommendation:
Due to the suggestions listed above, I would suggest that this manuscript need to be revised with minor revision before getting accepted.

Reviewer 3 ·

Basic reporting

No Comments

Experimental design

No Coments

Validity of the findings

No comments

Comments for the author

In this manuscript, the authors carried out behavioral, immunohistochemical and western blot experiments to show that social isolation caused cognitive deficits, along with increased expression of ADAR1 in selective areas such as hippocampus and frontal cortex. Overall the study itself seems to be solid. However, there is a list of issues with the way the results are currently presented. The authors need to address these issues before the study can be published.
1. Although the text was written in understandable English, it still reads awkward and may sometimes cause confusion. I highly recommend that the authors seek help from professional editing services or native English speakers to edit it before sending back for review.
2. Fig 2 A and B should be combined because a large part of Fig2B is re-plotting the data already shown in A. The symbols used in figures should be consistent throughout the manuscript.
3. Fig 3 A needs to be simplified to two panels, each showing the area of hippo or frontal cortex.
4. Fig 3B,C need to be reorganized with C1 and C2 as a single row on top, followed by C2W, SI2W, SI2WR as second row, C4W, SI4W as the 3rd row, and C8W and SI8W as the bottom row.
4. In Fig 3C, the double staining in CA1 is not a good choice because the nissel stain covers ADAR1. Single ADAR1 staining and comparison of optical density between different treatment groups seems a better solution.
5. In Fig 5, there are indications that some of the ADAR1 bands were individually cut and put together. This practice should be highly discouraged. Please show the results from a single experiment.
Fig 7 does not provide any thing new, and should be removed.
6. In the title and text, the authors need to pay attention to use the correct words to describe their findings more precisely. For example, The title reads '...leading to...'. However, the authors did not present any data that can support this causal relationship. All the data presented in this manuscript are correlative rather than causal.

---

## Round 0.2 · accepted · Accept

· Academic Editor

Accept

The authors have substantially revised the manuscript. I am pleased to inform you that the manuscript is acceptable for publication in PeerJ pending a very minor comment from reviewer #3 (which can be edited while in production).

·

Basic reporting

The authors have revised the manuscript and answered my questions. It is suitable for acceptance.

Experimental design

The authors have revised the manuscript and answered my questions. It is suitable for acceptance.

Validity of the findings

The authors have revised the manuscript and answered my questions. It is suitable for acceptance.

Reviewer 2 ·

Basic reporting

I accept to publish the revised manuscript based on their revisions.

Experimental design

I accept to publish the revised manuscript based on their revisions.

Validity of the findings

I accept to publish the revised manuscript based on their revisions.

Comments for the author

I accept to publish the revised manuscript based on their revisions.

Reviewer 3 ·

Basic reporting

No Comments

Experimental design

No Comments

Validity of the findings

No Comments

Comments for the author

The authors have done a great job revising the manuscript. I recommend accepting this manuscript with only one very minor suggestion:

The authors wrote 'To prove our hypothesis, ...' in the abstract. Please remember that a hypothesis can be tested and disproved, but it can never be 'proved'. The experimental results, no matter how solid or definitive they are, can only support a hypothesis, but they can never prove (to be true) it. Please revise the text to 'To test our hypothesis,...'.

Congratulations